# Anti-drift pose tracker (ADPT), a transformer-based network for robust animal pose estimation cross-species

Guoling Tang[1,2†], Yaning Han[1,2†], Xing Sun[1,2], Ruonan Zhang[3], Ming-Hu Han[1,4], Quanying Liu[5]*, Pengfei Wei[1,2]*

[1]University of Chinese Academy of Sciences, Shenzhen, China; [2]University of Chinese Academy of Sciences, Beijing, China; [3]Guangxi University of Science and Technology, Liuzhou, China; [4]Shenzhen University of Advanced Technology, Shenzhen, China; [5]Department of Biomedical Engineering, Southern University of Science and Technology, Shenzhen, China

*For correspondence:
liuqy@sustech.edu.cn (QL);
pf.wei@siat.ac.cn (PW)

†These authors contributed equally to this work

## eLife Assessment

This **useful** study introduces a deep learning-based algorithm that tracks animal postures with reduced drift by incorporating transformers for more robust keypoint detection. The efficacy of this new algorithm for single-animal pose estimation was demonstrated through comparisons with two popular algorithms. The strength of evidence is **solid** but would benefit from consideration of issues in multi-animal tracking. This work will be of interest to those interested in animal behavior tracking.

**Abstract** Deep learning-based methods have advanced animal pose estimation, enhancing accuracy, and efficiency in quantifying animal behavior. However, these methods frequently experience tracking drift, where noise-induced jumps in body point estimates compromise reliability. Here, we present the anti-drift pose tracker (ADPT), a transformer-based tool that mitigates tracking drift in behavioral analysis. Extensive experiments across cross-species datasets—including proprietary mouse and monkey recordings and public *Drosophila* and macaque datasets—demonstrate that ADPT significantly reduces drift and surpasses existing models like DeepLabCut and SLEAP in accuracy. Moreover, ADPT achieved 93.16% identification accuracy for 10 unmarked mice and 90.36% accuracy for freely interacting unmarked mice, which can be further refined to 99.72%, enhancing both anti-drift performance and pose estimation accuracy in social interactions. With its end-to-end design, ADPT is computationally efficient and suitable for real-time analysis, offering a robust solution for reproducible animal behavior studies. The ADPT code is available at https://github.com/tangguoling/ADPT.

## Introduction

Animal behavior is a complex and dynamic phenomenon that is shaped by a wide range of factors, including environment, genetics, diseases, cognitive states, and social interactions (*Robinson et al., 2008*). Understanding the underlying mechanisms and neural correlates of animal behaviors requires accurate and detailed pose tracking as they move freely (*Pereira et al., 2020*; *Krakauer et al., 2017*). Recently, deep learning-based tools such as DeepLabCut, SLEAP, and DeepPoseKit have offered the feasibility of automatically quantifying complex freely moving animal behaviors from videos recorded by contactless cameras (*Mathis et al., 2018*; *Pereira et al., 2022*; *Graving et al., 2019*). Nevertheless, these deep learning methods are susceptible to uncertainty and noise interference, leading

to tracking drift due to errors in the detection of one or more keypoints, in the estimated keypoint dynamics (*Weinreb et al., 2024*; *Hsu and Yttri, 2021*; *Lonini et al., 2022*). Such drift in keypoints estimates can broadly affect subsequent animal behavior statistics and downstream tasks, such as behavior classification, individual identification, and social behavior clustering (*Sheppard et al., 2022*; *Huang et al., 2021*). It severely jeopardizes the reliability and repeatability of ethological studies. Thus, there is an urgent need for an anti-drift pose tracking tool for animal behavior analysis.

Tracking drift of pose estimation, occurring at the upstream behavioral analysis, generally hinders all downstream behavior-related studies. For example, animal gait analysis relies on accurate tracking of limbs and paws (*Sheppard et al., 2022*), and behavioral classification relies on the dynamics of body keypoints (*Huang et al., 2021*; *Han et al., 2022*). So far, deep learning pose estimation has not achieved the reliability of classical kinematic analysis, which often involves post-processing in real-world applications (*Niknejad et al., 2023*; *Aljovic et al., 2022*; *Weinreb et al., 2024*). One major reason is tracking drift. The drifted keypoints may be unsystematically distributed within each predefined behavior class, misleading the decision boundaries of the behavior class, thereby reducing the performance of supervised behavior classification *Gabriel et al., 2022* or unsupervised behavior representation (*Huang et al., 2021*). Even the state-of-the-art (SOTA) deep learning methods such as DeepLabCut and SLEAP have no effective strategies to avoid the tracking drift (*Weinreb et al., 2024*; *Mathis et al., 2018*; *Pereira et al., 2022*; *Graving et al., 2019*; *Lauer et al., 2022*). Inherited from the tracking drifts, the inaccuracy of pose estimation, gait analysis, and behavioral classification may result in wrong behavioral discoveries, such as those investigating behavioral correlates of genes, neural circuits, and neuropsychiatric diseases (*Sheppard et al., 2022*; *Huang et al., 2021*; *Liu et al., 2021*; *Han et al., 2022*). These concerns, along with issues related to the safety of deep learning tools, have slowed the widespread application of deep learning-based methods in behavioral analysis and limited the development of ethology.

There are three strategies to eliminate tracking drift in current SOTA methods of animal pose estimation. The first strategy is human refinement or human in the loop (*Mathis et al., 2018*; *Pereira et al., 2022*). DeepLabCut (DLC) and SLEAP both embed a user interface to allow humans to exclude and rectify outliers frame by frame (*Mathis et al., 2018*; *Pereira et al., 2022*). Although it would be the golden criterion to reduce the tracking drift, this strategy restricts the efficiency of the biological experiment when the human faces millions of drifted frames. The second strategy is signal processing filters such as median filter and low pass filter (*Stenum et al., 2021*; *Pereira et al., 2019*; *Luxem et al., 2022*; *Weinreb et al., 2024*; *Han et al., 2024*; *Li and Lee, 2021*). They can efficiently remove most of the drifted points without human intervention, but they will also remove the subtle behaviors with high-frequency features such as self-grooming in autism mouse models (*Huang et al., 2021*) or tremor in animal models of Parkinson's disease (*Baker et al., 2022*). The third strategy is fitting the drifted frames using linear dynamic models such as Keypoint-Moseq (*Weinreb et al., 2024*) and adaptive Kalman filter (*Huang et al., 2022*). They can reduce the drift and maintain the high-frequency behaviors at the same time. Nevertheless, the performance of these models would drop sharply when processing continuous and long-duration drifted frames. These three strategies are only expedient to reduce tracking drift after pose estimation, whose performances are also restricted by the tracking accuracy of raw frames. Therefore, the elimination of tracking drift should be tackled from the beginning of the deep learning pose estimation step.

The structure design of the artificial neural network (ANN) is the first step to correct tracking drift. DeepLabCut, SLEAP, and DeepPoseKit all take the convolutional neural network (CNN) as the main component of pose estimation ANNs, which is the core problem causing tracking drift (*Mathis et al., 2018*; *Pereira et al., 2022*; *Graving et al., 2019*; *Lauer et al., 2022*). The limited working memory of the CNN makes it easy to be influenced by the content-independent parameters to predict the wrong locations of keypoints and finally cause tracking drift (*Yang et al., 2021*). To avoid this drawback, the Transformer becomes a better option to construct pose estimation ANNs because it is more efficient to capture global dependent features of images (*Yang et al., 2021*; *Stoffl et al., 2021*; *Xu and Zhang, 2022*). Although Transformer-based ANNs have achieved new SOTA in lots of human pose estimation datasets, it is rarely applied in animal pose estimation. Different from human poses, animal poses have more indistinct body structures because they are covered by furs (*Vidal et al., 2021*). In addition, the well-annotated animal pose datasets are not abundant enough to cover various experiment settings. Experimenters always need to make customized datasets for their specific applications

(*Han et al., 2024*). Therefore, the application of the Transformer to reduce tracking drift in the animal pose estimation task still needs an elaborate design of ANN structures.

To import the Transformer to overcome the tracking drift of animals, we designed an ADPT following the characteristics of animal behavior data. CNN and Transformer are cascaded with skip connections to capture subtle animal appearance features from only hundreds of labeled frames (*He et al., 2016*; *LeCun et al., 2015*; *Vaswani et al., 2017*). This structure design makes ADPT show significantly fewer tracking drifts than (*Mathis et al., 2018*; *Pereira et al., 2022*). The effect of anti-drift of ADPT is universally validated in the public datasets and our customized datasets including *Drosophilas*, mice, and macaques, which demonstrates that ADPT is robust in broad application scenarios cross-species (*Pereira et al., 2019*; *Bala et al., 2020*; *Han et al., 2024*). ADPT also achieves robust pose estimation and identity recognition of free-interactive mice combined with a mix-up dataset generation strategy. The results of markerless identity recognition show that the feature extraction of ADPT is reliable enough to cover both multi-animal pose estimation and identity recognition tasks, which are more difficult than the single-animal pose estimation task (*Agezo and Berman, 2022*; *Lauer et al., 2022*; *Han et al., 2024*). It reduces the computational time cost and increases the throughput of behavioral data processing because ADPT does not need a multi-stage neural network such as (SIPEC *Marks et al., 2022* or Social Behavior Atlas *Han et al., 2024*). Together, ADPT would be an accessible tool to reduce the pose tracking shift across species from the upstream of behavior analysis. ADPT has the potential to improve the reliability of computational ethology-based biological studies.

## Results

### Anti-drift pose tracker

Existing deep learning-based methods often produce some unreliable pose estimation, such as interference caused by similar objects, keypoint drift, and failures of body part detection (*Figure 1A*). To clarify these errors, we use 'track' or 'tracking' to refer to the tracking of all body points or poses of an individual, and 'detect' or 'detection' when referring to specific keypoints. These estimation errors largely compromise the robustness of pose estimation in freely behaving animals, which can affect the statistical results of behavioral analyses and sometimes even lead to erroneous scientific findings. In this study, we present a reliable animal behavioral analysis tool, called the ADPT. ADPT can effectively eliminate estimated drifts (*Figure 1B*). ADPT is a heatmap-based pose estimation network that inferences input images to confidence heatmap, location refinement, and low-resolution semantic segmentation (LRSS) (*Figure 1C*). In the network architecture of ADPT (*Figure 1D*), we utilize the convolutional structure to extract local information on the one hand, and the transformer attention mechanism to learn the long-term global dependencies on the other hand. Compared with purely attention-based network structures (such as ViT *Yang et al., 2021*; *Stoffl et al., 2021*; *Xu and Zhang, 2022*), our CNN-transformer structure can significantly reduces the number of model parameters and, therefore, requires fewer training data samples. It is particularly suitable for data-limited applications such as animal behavior analysis.

### Customized behavioral videos for testing ADPT

The identification of drifting keypoints relies heavily on videos generated during inference or visualized coordinates. Yet there is no publicly available video dataset specifically designed for anti-drift evaluation. To fill this gap, we collected behavioral data from mice and monkeys (see *Figure 3—video 1*, *Figure 4—video 1*). For single animal pose estimation, we recorded videos from free-moving mice and monkeys with four cameras and then hand-labeled randomly extracted frames. For mice, we labeled these frames with 16 keypoints, including nose, eyes, ears, front limbs, front claws, back, hind limbs, hind claws, root tail, mid tail, and tip tail. For monkeys, we labeled these frames with 17 keypoints, including nose, eyes, ears, shoulders, elbows, hands, hips, knees, and ankle. Given the popularity of mouse behavioral study, mice served as our primary subjects for evaluation, with videos obtained from four different perspectives involving four distinct individuals. Each mouse video spanned 15 min. The training dataset comprised 440 randomly extracted images from these videos and other collected videos (training:validation = 95%:5%). Monkey videos, on the other hand, encompassed eight different viewpoints, featuring multiple individuals, from which a 30 min video was used for performance evaluation. The training set consisted of 3488 randomly sampled images

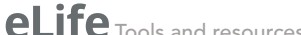

**Figure 1.** Anti-drift pose tracker (ADPT). (**A**) Three examples of drifts in deep learning-based animal behavioral analysis. Similar object disturbance means that the object similar to a specific body part misleads the deep learning-based methods. Inexplicable keypoint drift is caused by the high confidence score predicted on the wrong place by the network. Failure to detect the keypoint is probably caused by the predicted low confidence score. (**B**) The anti-drift effects of ADPT. (**C**) The general workflow of ADPT. The network is trained to predict confidence heatmap, low-resolution semantic segmentation (LRSS), and location refinement. (**D**) The network architecture of ADPT.

(training:validation = 95%:5%). For social or multi-animal pose estimation, we recorded 10 min videos of freely socializing mice in a homecage from three different perspectives. We manually labeled 1200 images for training and validation (training = 95%:5%) and also labeled the back locations of two mice during the first minute to evaluate tracking accuracy. Using our dataset, we trained ADPT,

DeepLabCut, and SLEAP models, separately, to detect body keypoints from behavioral videos. The behavioral data is available via https://github.com/tangguoling/ADPT/blob/main/data/link.md.

## ADPT demonstrates the remarkable anti-drift performance

Firstly, we visualized the time course of seventeen estimated key body parts from a 1 min segment of mouse videos (*Figure 2A*), demonstrating the anti-drift effects of ADPT. In contrast, the other two deep learning-based methods suffer from drift and misses of body parts. Then, we zoomed into the frames of failures in *Figure 2B*. The quantitative results of 240 min videos from two mice were shown in *Figure 2C and D*. Interestingly, DeepLabCut has almost the same probability of generating drift and misses, while SLEAP was more prone to misses. As presented in *Figure 2D*, the tip tail was the most challenging part of the body for both drifts and missed due to the long distance from the tip tail to the rest of the body. For CNN-based methods such as DeepLabCut and SLEAP, learning such long-range tail-body relationships is particularly difficult, while the attention mechanism of ADPT allows it to learn long-range dependencies. Due to frequent occlusion in the video, the left and right claws could be easily missed or drifted. Our model evaluations show that ADPT has significantly lower root mean squared errors compared to SLEAP and achieves comparable or improved accuracy compared to DeepLabCut (*Figure 2E*), suggesting that ADPT can reliably detect the hind claws, offering a potential tool for gait analysis and tail-related behavior paradigms.

## Anti-drift performance remains consistent irrespective of the video background and individual animals

Any measuring tool that exhibits biased measurement errors towards specific subjects introduces inaccuracies in its assessments. For example, if the model accurately estimates the posture of mouse A but experiences greater posture drift in estimating mouse B, this discrepancy leads to measurement errors, impacting subsequent behavioral analyses. Hence, to evaluate the independence of posture estimation's anti-drift effect concerning individual animals or background factors, we conducted one-way ANOVA on the tracking results. We trained ADPT, DeepLabCut, and SLEAP five times each and applied these models to infer behavioral videos across different individuals and video backgrounds. Firstly, we compared ADPT's anti-drift performance across different individuals and backgrounds. The results showed that ADPT exhibited significantly lower drift percentages than the other two methods across different individuals and video backgrounds (*Figure 3A and C*). Then, the inference results were grouped based on individual animals and video backgrounds, respectively, for five individual one-way ANOVA analyses. The results of these five ANOVA analyses are presented in *Figure 3B and D*. Our analyses revealed that drift occurrences were more significantly affected by backgrounds in DeepLabCut, while individual variations had a more significant impact on SLEAP. However, ADPT showed slight susceptibility to background influence. Consequently, we assert that in comparison to DeepLabCut and SLEAP, ADPT only demonstrates a lesser susceptibility to the influence of individual animals and background factors. This resilience significantly mitigates biases in tracking results. ADPT's ability to generate fewer biases due to individual or background factors during inference holds promise for achieving better consistency in downstream behavioral analyses. This analysis also underscores the importance, when using ADPT, of minimizing background variations, ideally maintaining consistent backgrounds.

## Cross-species anti-drift capability of ADPT is reliable

While ADPT has demonstrated exceptional anti-drift abilities in mice, numerous other animal models are employed in behavioral studies. To validate the robustness of ADPT in tracking different species, particularly those posing significant tracking challenges, we selected cynomolgus monkey as a species known for its complexities in tracking. We utilized the models to track a video in which both humans and monkeys appeared simultaneously, presenting similar objects in the scene. Visualizing the keypoint tracking results from 1 min time course featuring both entities allowed us to showcase the anti-drift efficacy of ADPT (*Figure 4A*). In contrast, the other two methods exhibited tracking failures when humans were present, as illustrated in the zoomed-in frames of failure in *Figure 4B*. When humans were present, both DeepLabCut and SLEAP exhibited instances of tracking drift, whereas ADPT remained unaffected by the presence of similar objects. Similarly, we evaluated the performance of 'drift' and 'miss' for various body parts in this scenario. We observed that ADPT consistently

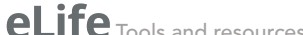

**Figure 2.** Analysis of anti-drift pose trackers (ADPT's) anti-drift performance in a mouse dataset collected by our lab. (**A**) The time course of the y-axis position of sixteen body parts extracted from a 1 min video using ADPT, DeepLabCut, and SLEAP tools. It showed that ADPT successfully detected all 16 body parts of a mouse, whereas DeepLabCut and SLEAP encountered inexplicable tracking drifts. (**B**) Two anti-drift examples from ADPT, where the tail was drifted by DeepLabCut and the hind claw failed to detect by SLEAP. (**C**) Overall percentage of tracking drift and failing to detect (miss) frames from three methods. ADPT demonstrated a significantly lower drift percentage than other methods. (**D**) The percentage of frames with tracking drift

*Figure 2 continued on next page*

*Figure 2 continued*

(left) and failing to detect (right). Drifts were mainly from the top four body parts, including the tip tail, the left and the right hind claws, and the middle tail. (**E**) The averaged RMSE across all body parts (left) and RMSE of the top four body parts with drifts (right). ADPT achieved the smallest RMSE than other two tools when thresholded at 0.2. *p<0.05, **p<0.01, ***p<0.001, ****p<0.0001. RMSE: root mean square error.

**Figure 3.** Anti-drift performance cross background and individual, where the percentage of frames includes two types of drift phenomena: drift and miss. (**A**) The overall cross-individual anti-drift performance of anti-drift pose tracker (ADPT) and the other methods. The drift percentage of ADPT is significant lower than other methods. (**B**) After training the model 5 times on the dataset shuffle, the cross-individual drift percentage for each shuffle was analysed using one-way ANOVA. The ANOVA results revealed that there are differences in the inference results of the SLEAP model among individual, and there were no differences for ADPT or DeepLabCut. (**C**) The overall cross-background anti-drift performance of ADPT and the other methods. The drift percentage of ADPT is significant lower than other methods. (**D**) The cross-background drift percentage for each shuffle was analysed using one-way ANOVA. The ANOVA results revealed that there are slight differences in the inference results of the DeepLabCut model among individual, and there were no differences for ADPT or SLEAP. ns.: no significant, *p<0.05, **p<0.01, ***p<0.001, ****p<0.0001.

The online version of this article includes the following video for figure 3:

**Figure 3—video 1.** Video file containing clips of mouse behavior videos.
https://elifesciences.org/articles/95709/figures#fig3video1

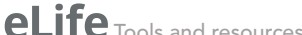

**Figure 4.** Analysis of anti-drift pose trackers (ADPT's) anti-drift performance on monkey data, showing the cross species anti-drift ability. (**A**) The time course of the $y$-axis position of sixteen body parts extracted from a 1 min video using ADPT, DeepLabCut, and SLEAP tools. It showed that ADPT successfully detected all 17 body parts of a monkey, while the other two methods encountered tracking drift because of the appearance of humans. (**B**) DeepLabCut and SLEAP both mistakenly located the monkey's eyes on humans when they appeared, while ADPT can achieve robust tracking. (**C**, **D**) The percentage of frames with tracking drift and failing to detect (miss). The occurrence of drift was mainly concentrated in the limbs, because the appearance of humans.

The online version of this article includes the following video for figure 4:

**Figure 4—video 1.** Video file containing a clip of monkey behavior video.

https://elifesciences.org/articles/95709/figures#fig4video1

outperformed the other two methods overall (*Figure 4C and D*). However, given the more complex experimental setup and animal movements, ADPT exhibited slight instances of drift and 'fail to detect' effects.

Consequently, our findings suggest that our approach demonstrates remarkable anti-drift performance, cross-individual and cross-view capabilities. Notably, our anti-drift performance was more pronounced in consistent experimental scenarios. Our experiments with monkeys substantiated our method's profound cross-species anti-drift capability, emphasizing its significance in behavioral studies involving diverse animal models.

## Public datasets confirm the outperformance of ADPT in precision and practicality

In adddition to evaluating ADPT's performance on behavioral study videos, we recognized the significance of image datasets as benchmarks for assessing pose estimation effectiveness. Thus, to comprehensively evaluate the generalizability of ADTP performance to animals in skeletal complexity and body size, and the background complexity of videos, we used two public datasets, a single fly dataset (*Figure 5A*; *Pereira et al., 2019*), and a macaque OMS_Datase (*Bala et al., 2020*). The single fly dataset contains 1500 annotated frames of 32-node skeleton fly. To ensure a fair comparison, we followed the same dataset split strategy and data augmentation strategy described in *Pereira et al., 2022*. The evaluation metric used was mean Average Precision (mAP), which measures the accuracy of keypoint localization for all body parts, following the protocol established in *Pereira et al., 2022*. On the other hand, The OMS_Dataset (*Bala et al., 2020*) is a large database of annotated macaque images (*Figure 5F*). To evaluate the performance of our methods, we randomly selected 5000 images out of 195,228 images from this dataset and resized them to 368×368 resolution. We split the dataset into 40% training data and 60% validation data. We employed the same strategy used in the default configuration of DeepLabCut toolbox to augment the data. The average distance (root square mean errors, RMSE) between the ground truth and predicted keypoints and the mAP were used as evaluation metrics. *Figure 5A and F* presented several examples annotated by ADPT on these two datasets, respectively. Furthermore, to verify the practicality of ADPT, we also evaluated the amount of required training data and the inference speed of the model. Finally, we evaluated the scalability of ADPT on the StanfordExtra dataset (*Biggs et al., 2020*). Our results demonstrated the capability of ADPT on non-laboratory dogs (*Figure 5—figure supplement 1* and *Figure 5—video 1*). These evaluations underscore ADPT's versatility, showcasing its robustness and accuracy in diverse animal contexts, thereby affirming its potential as a highly adaptable tool for comprehensive behavioral studies.

## ADPT offers higher tracking accuracy than existing SOTA methods

The tracking performance of ADPT was compared with the existing SOTA methods, such as DeepLabCut and SLEAP (*Figure 5B, G and H*). On the single-fly dataset, ADPT exceled with an average mAP of 92.83%, surpassing both DeepLabCut and SLEAP (*Figure 5B*). On the OMS Dataset, ADPT exhibited significant advantages in terms of mAP, RMSE (threshold = 0.2), RMSE (threshold = 0.6), achieving values of 30.9%, 8.32, and 6.25, which were significantly superior to SLEAP, and slightly outperforming DeepLabCut when the threshold set as 0.6 (*Figure 5G*, *Supplementary file 1*). Moreover, we further examined the detection of macaque hip and tail on OMS_Dataset (*Figure 5H*). We found that ADPT's tracking performance of tail is better than DeepLabCut and SLEAP, while the hip tracking is equivalent to DeepLabCut and better than SLEAP. This further demonstrates the superiority of ADPT in tail-related behavior paradigms. By conducting evaluations on these diverse datasets, we aimed to assess the robustness and generalizability of our methods across more different animal species, pose complexities, and environmental conditions. The results obtained from these evaluations provide solid proof of the performance and potential of our methods for single-animal pose estimation.

Since annotating behavioral data is tedious, a deep learning-based method that does not require large amounts of annotated data is crucial. Here, we studied how the accuracy of ADPT changes with the amount of annotated data. Notably, ADPT achieved acceptable performance using only 350 annotated images (*Figure 5D*), indicating that ADPT is data efficient.

**A**  Samples of prediction single fly dataset

**B**  Single-animal pose estimation
Dataset, single fly (threshold = 0.2)

**C**  Baseline vs Baseline + LRSS
Dataset, single fly

**D**  Single-animal pose estimation
Dataset, single fly

**E**  baseline vs baseline + transformer
Dataset, single fly

**F**  Samples of prediction on OMS_Dataset

**G**  Single-animal pose estimation
Dataset, OMS_Dataset

**H**  Single-animal pose estimation
Dataset, OMS_Dataset (threshold = 0.2)

**Figure 5.** Results of public datasets evaluation. (**A**) Samples of prediction on single fly dataset. (**B**) Mean average precision (mAP) on fly dataset, where anti-drift pose tracker (ADPT) achieved average of 92.8% accuracy (the best model achieved 93.27%). (**C**) Low-resolution semantic segmentation (LRSS) improved the average accuracy by 0.3% on a single fly dataset. (**D**) Relationship between annotated image and accuracy of ADPT on fly dataset where ADPT achieved acceptable performance with only 350 annotated images in a simple laboratory environment. Points indicate the validation accuracy

*Figure 5 continued on next page*

*Figure 5 continued*

of model training on specific number of labels dataset. (**E**) Transformer improved the average accuracy by 0.4% on a single fly dataset. (**F**) Samples of prediction on OMS_Dataset. (**G**) Root mean square error (RMSE) on OMS_Dataset, where ADPT achieved smaller root square mean error (RMSE) than SLEAP when threshold = 0.2, and smaller than DeepLabCut when threshold = 0.6. p-value, **: 0.001862, ns.: 0.243472, ***8.700e-06. (**H**) RMSE comparison on hip and tail of OMS_Dataset. p-value, ***0.000561, Hip ns.:0.023766, Tail ns.:0.336642, *: 0.035782.

The online version of this article includes the following video and figure supplement(s) for figure 5:

**Figure supplement 1.** Picture examples of dog pose estimation.

**Figure 5—video 1.** Video examples of dog pose estimation.

https://elifesciences.org/articles/95709/figures#fig5video1

## ADPT's fast inference enables real-time applications

Here, we evaluate the inference speed of ADPT. We compared it with DeepLabCut and SLEAP on mouse videos at 1288x964 resolution. Our method exhibited an impressive prediction speed of 90±4 frames per second (fps), faster than DeepLabCut (44±2 fps) and equivalent to SLEAP (106±4 fps). These results highlighted the efficient inference capabilities of ADPT, which is crucial for real-time applications and the analysis of large-scale behavioral data.

## LRSS and transformer help improve tracking accuracy

To examine the contribution of the low-resolution semantic segmentation (LRSS) and the transformer architecture to ADPT, we conducted two ablation studies using the fly dataset. We compared multiple variants to uncover the impacts of the LRSS module and the transformer module on pose estimation performance. First, we explored the influence of LRSS by comparing the performance of the complete ADPT with the one removed LRSS. As shown in *Figure 5C*, LRSS module can improve the average accuracy by 0.2%. Moreover, to assess the role of transformer architecture, we conducted a comparative analysis between the complete ADPT with the transformer and a variant of the model where the transformer was removed. As shown in *Figure 5E*, the transformer improved the average accuracy by 0.4%, suggesting the benefits of the transformer architecture in pose estimation.

## ADPT can accurately track the non-laboratory dog

To test the generalizability of our approach beyond laboratory-behavior animals, we applied ADPT to the keypoint detection task for the non-laboratory dog. The dataset is from *Biggs et al., 2020*. We randomly divided the dataset into 85% and 15% training and validation data. ADPT was instantiated with the same network architecture for laboratory animal pose estimation, showcasing the versatility of ADPT. We followed *Biggs et al., 2020* and used Percentage of Correct Keypoints (PCK) metric to evaluate the accuracy of keypoint detection. The results showed that ADPT achieved an average 86.54% PCK score (legs: 85.54%, tail: 79.89%, ears: 88.61%, face: 95%). Examples of identified keypoints of dogs were shown in *Figure 5—figure supplement 1*; *Figure 5—video 1*. These results supported the flexibility of ADPT in different animal species and potentially more challenging real-world scenarios.

## ADPT can be adapted for end-to-end pose estimation and identification of freely social animals

We adapted ADPT to end-to-end tracking of the social interacting mice with similar appearances. To this end, we added additional heads after feature concatenation and utilized LRSS to confirm the identities of the mice. We generated a multi-animal dataset for social tracking by mixing up two labeled frames from single mouse videos (*Figure 6*). The evaluation of our social tracking capability was performed by visualizing the predicted video data (see *Figure 7—video 1* and *Figure 7—video 2*).

Prior to social tracking, we evaluated identity-tracking accuracy using a dataset consisting of 10 mouse videos of different individuals. The overall workflow of these extended applications is depicted in *Figure 7*. Initially, we utilized a variant of ADPT (empowering LRSS with identity information) for simultaneous animal pose estimation and identity synchronized tracking. For each frame, identity recognition was based on the LRSS output by ADPT (*Figure 7A*). Although the appearance of the mice is very similar, our experimental results showcased a remarkable 93.16% accuracy in identity

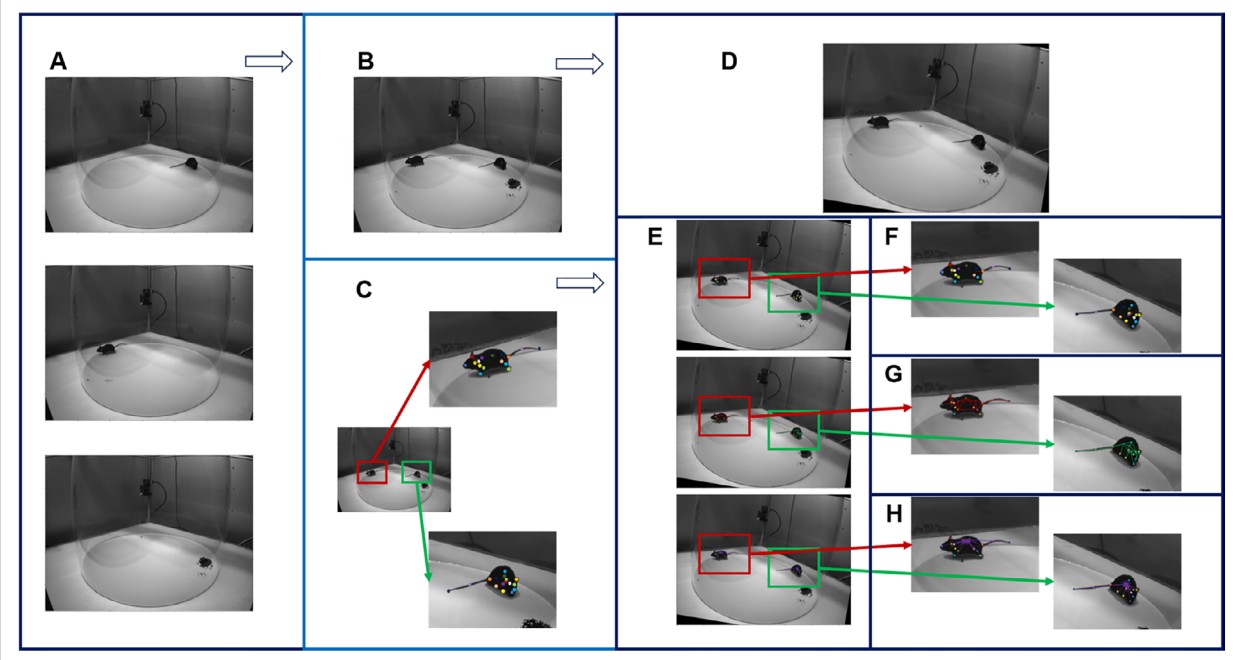

**Figure 6.** Illustration for mix-up social animal dataset generation. (**A**) Frames originating from different videos and corresponding background. (**B**) Mix-up image. (**C**) Represents schematic diagrams illustrating the keypoint generated from single animal pose estimation of anti-drift pose tracker (ADPT). (**D**) Represents an augmented mix-up image. (**E**) Represents schematic diagrams of augmented annotation. (**F**) Represents augmented keypoints. (**G**) Represents augmented low-resolution semantic segmentation (LRSS). (**H**) Represents schematic diagrams of augmented Body Affinity Fields (BAF), inspired by Part Affinity Fileds (*Cao et al., 2021*).

recognition (*Figure 7B*). This approach demonstrates LRSS's capability to record individual identities like semantic segmentation masks. The outcomes, showcased in *Figure 7—video 1*, manifested synchronized tracking of identity and pose estimation.

Subsequently, we tested the tracking performance with free-social animals. Inspired by Part Affinity Fileds (*Cao et al., 2021*), we created Body Affinity Fileds (BAF) to help distinguish different individuals. BAF and LRSS were used together to identify individuals. In the first scenario, We trained ADPT on the Mix-up social animal dataset and employed it to predict 1 min free-social video of mice with similar appearance. Without additional temporal post-processing, ADPT achieved a 90.36% accuracy in identity recognition, as referenced in *Figure 7—video 2*. Following temporal identity correction, ADPT remarkably achieved a 99.72% accuracy in identity recognition (*Figure 7C*), as shown in *Figure 7—video 2*. The pose estimation accuracy was acceptable, but we recognized that there are detection errors of tail or tracking errors when animals are very closed sometimes which may be due to the lack of real-world training data.

In the second scenario, we trained ADPT on manual labeled homecage social mice dataset, a set of real-world training data (*Figure 8A*.) and used it to predict a 1 min free-social video. We evaluated anti-drift performance and found that ADPT outperformed Deeplabcut and SLEAP, achieving 15% improvement of pose estimation accuracy (*Figure 8D and E*) and almost 5–10 times improvement of tracking accuracy (*Figure 8C*). *Figure 8B* illustrates ADPT's prediction of the x-coordinates of different mice, demonstrating less keypoint drift. In *Figure 8C*, we compare the anti-drift performance of the raw predictions from the three methods, highlighting ADPT's superior tracking performance compared to DeepLabCut and SLEAP. Furthermore, we assessed pose estimation accuracy between ADPT and DeepLabCut/SLEAP, showing that ADPT has better pose estimation accuracy than both SLEAP and DLC (*Figure 8D and E*). Lastly, an ablation study confirmed that BAF improves pose estimation accuracy (*Figure 8F*). The tracking result of this scenario was shown in *Figure 8—video 1*.

In addition to mice, we evaluated the pose estimation accuracy of ADPT on the marmoset dataset, a publicly available resource (*Mathis et al., 2018*). We adhered to the default marmoset configuration of DeepLabCut, randomly dividing the dataset into training and validation sets while employing the same data augmentation strategy. Under the evaluation metrics used by DeepLabCut, ADPT achieved

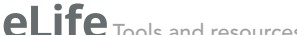

**Figure 7.** Applications of anti-drift pose tracker (ADPT) for multi-animal pose tracking. (**A**) Left: The pipeline for the multi-animal identity-pose tracking task. (**B**) Confusion matrix of the 10 mice classification (accuracy = 93.16%). (**C**) Social mice tracking pipeline with identification accuracy of 99.72%.

The online version of this article includes the following video(s) for figure 7:

**Figure 7—video 1.** Video file demonstrating single animal pose estimation and identity synchronized tracking.
https://elifesciences.org/articles/95709/figures#fig7video1

**Figure 7—video 2.** Video file demonstrating social animal pose estimation and identity synchronized tracking.
https://elifesciences.org/articles/95709/figures#fig7video2

an average accuracy of 6.14±0.19 pixels, whereas DeepLabCut reached 6.63±0.09 pixels. Additionally, when assessed using SLEAP evaluation metrics, ADPT achieved an average accuracy of 7.02±0.11 pixels, compared to SLEAP's 11.39±0.31 pixels.

Together, these different applications demonstrate the versatility of ADPT, ranging from single animal pose estimation to complex situations involving social interactions. ADPT's versatility and adaptability paves the way for comprehensive behavioral studies.

## Discussion

Here, we have presented ADPT, a transformer-based pose tracker, to address the pose drift problem in animal pose estimation. The core of ADPT is the elaborate combination of the convolutional network *LeCun et al., 2015* and transformer layers (*Vaswani et al., 2017*), with the goal of capturing

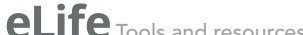

**Figure 8.** Evaluation of anti-drift pose tracker (ADPT) for homecage social mice scenario. (**A**) Illustration of homecage social mice dataset. (**B**) Filtered predicted back locations of different mice by ADPT. (**C**) Comparison of different methods and manual labels. We trained each model three times, and this figure presents the results from one of those training sessions. We calculated the average root square mean error (RMSE) between predictions and manual labels, demonstrating that ADPT achieved an average RMSE of 15.8±0.59 pixels, while DeepLabCut (DLC) and SLEAP recorded RMSEs of 113.19±42.75 pixels and 94.76±1.95 pixels, respectively. (**D**) Pose estimation accuracy comparison between ADPT and DLC based on the DLC evaluation metric. ADPT achieved an accuracy of 6.35±0.14 pixels across all body parts of the mice, while DLC reached 7.49±0.2 pixels. (**E**) Pose estimation accuracy comparison between ADPT and SLEAP using the SLEAP evaluation metric. ADPT achieved 8.33±0.19 pixels across all body parts of the mice, compared to SLEAP's 9.82±0.57 pixels. (**F**) Body affinity fields (BAF) improved pose estimation accuracy by 0.4 pixels under the SLEAP evaluation metric.

The online version of this article includes the following video for figure 8:

**Figure 8—video 1.** Video file demonstrating homecage social mice pose estimation and identity synchronized tracking.
https://elifesciences.org/articles/95709/figures#fig8video1

both local details and global context. This architecture helps ADPT achieve a more reliable feature extraction on animal objects, resulting in higher accuracy in tracking the poses frame by frame with less drifts or misses, compared to *Lauer et al., 2022*; *Pereira et al., 2022*. In addition, we presented the procedure for the data generation of Mix-up social animals, which is convenient and effective for exponentially synthesizing new data to improve the performance of ADPT. We showed that ADPT can be used for multi-animal pose estimation and identification. These two tasks were considered much more difficult than single-animal pose estimation (*Lauer et al., 2022*). The end-to-end network structure of ADPT only needs to calculate one model loss so it is more computationally efficient than the multi-stage methods such as SIPEC and Social Behavior Atlas (*Marks et al., 2022*; *Han et al., 2024*). These advances show that ADPT is an accurate, universal, and efficient method, suggesting broader

application scenarios in neuroscience, genetics, and drug discovery. Now, the toolbox of ADPT has also been released at (*Tang and Sun, 2025*).

As the higher resolution of microscopy promotes the discovery of biological microstructures, the higher precision of animal pose estimation helps to detect subtle behavior structures and patterns, advancing ethology research. Behavior structures have been proven to be the signatures, fingerprints, and biomarkers to indicate disease developments (*Bohic et al., 2023*; *Gschwind et al., 2023*), genetic mutations (*Liu et al., 2021*; *Huang et al., 2021*; *Han et al., 2024*), and drug effects (*Wiltschko et al., 2020*; *Han et al., 2024*). Although these studies refine the behavior to module level (*Wiltschko et al., 2015*), this spatiotemporal scale of behavior structures is not sufficient to support finer animal studies such as decoding millisecond neural recordings with un-drifted poses (*Schneider et al., 2023*). Therefore, improving the accuracy and reliability of animal pose estimation is of high need for behavioral studies. ADPT provides such a tool for animal pose estimation.

ADPT enables a wide range of downstream applications, for instance, aligning behavioral manifold from keypoint dynamics with the neural manifold from large-scale neural recordings (*Urai et al., 2022*). Recent advances in neural decoding of speech *Li et al., 2023*; *Metzger et al., 2023* and vision *Schneider et al., 2023*; *Takagi and Nishimoto, 2023* have achieved incredible performance, but the accurate neural decoding of poses is still an existing problem. ADPT can quantify the poses of animals to reach a high resolution like the microphone for speech acquisition or visual pixels, which is an improvement from the aspect of behavior data acquisition. The second application is the gait analysis for 3D movements. Non-human primates are not restricted to moving on the ground, and the 3D gait would reflect their abnormal state after modeling treatment (*Liang et al., 2023*; *Thota and Alberts, 2013*). ADPT decreases the pose drift caused by body occlusion of single-view frames, which would reduce the error of 3D gait reconstruction. It also reduces the number of cameras for view angle compensation except for the profound understanding of 3D gait-related disorders (*Bala et al., 2020*). The third application is behavior-based drug screening (*Wiltschko et al., 2020*). Although MoSeq has built up the relationship between behavior syllables and psychoactive drugs (*Wiltschko et al., 2015*; *Wiltschko et al., 2020*), the resolution of behavior only exists at the syllable level. It is predictable for ADPT to improve the behavior resolution of MoSeq even Keypoint-MoSeq to a finer level to be not limited to the screen of psychoactive drugs (*Wiltschko et al., 2015*; *Weinreb et al., 2024*). In summary, solving the anti-drift problem from the very beginning of ADPT determines that it has widespread applications.

One potential improvement of ADPT is the design of positional encoding. With the increase in image size, the positional encoding would occupy more memory of the graphics processing unit. The process of high-resolution videos has to resize the frame to avoid being out of memory, in which the pixel-level information could be missed. Conditional positional encoding would be a possible solution to improve ADPT to face high-resolution frames (*Chu et al., 2021*). Another improvement of ADPT is using a more powerful backbone neural network. To facilitate the comparison between ADPT with other methods, the ResNet50 is used in all of the validation (*He et al., 2016*). Recent advances in the backbone such as (*Xu et al., 2022*) could be the better choice to replace ResNet and improve the performance of ADPT.

## Materials and methods

In this section, we first present ADPT method, then introduce the datasets used in each experiment, and finally describe the details of multi-animal experiments.

### The details of ADPT

Here, we present the key components and details of ADPT. We also provide the code for ADPT at https://github.com/tangguoling/ADPT/tree/main/code (*Tang, 2025*).

### The network architecture

Applying transformer in freely behaving animal pose estimation can help us alleviate keypoint tracking drift. Thus, we created a heatmap-based pose estimation model, called ADPT. The overall structure of the method and network is illustrated in *Figure 1C and D*. Initially, ADPT will resize image to a scale (a hyper parameter, the same as global_scale in DLC). Then, the network employs the stack1-2 of the

ResNet50 model to extract shallow-level features from the input images. At this stage, the images are extracted into features with a size of one-fourth of their original dimensions. Subsequently, network separately process these features in three branches, compute features at scale of one-fourth, one-eight and one-sixteenth, and generate one-eight scale features using convolution layer or deconvolution layer. Of particular significance is the utilization of the one-sixteenth scale feature, which is input into a transformer module for computation. This large-scale feature's involvement in the multi-head attention mechanism substantially enhances the model's ability to capture global relationships within the data. Finally, model concatenates these features by skip connections and compute them using convolution layers to generate output, including keypoint position confidence heatmaps, location refinement maps, low-resolution semantic segmentation map, and body affinity fields map.

### Low resolution semantic segmentation

In addition to generating the animal's skeletal keypoints, we also create a low-resolution semantic segmentation map (LRSS) of the animal. This segmentation map captures the coarse-level information about the different body parts or regions of the animal. By connecting the skeletal keypoints, the model can infer the boundaries, shapes, and identities of these regions. According to keypoints set *kps* of all individuals in frame, the pixel p-value at segmentation map is defined as,

$$M(p) = \begin{cases} identity, & \text{if } p \text{ on } limb_{ij}, \text{for } i, j \text{ in } kps \\ 0, & \text{otherwise} \end{cases} \quad (1)$$

The low-resolution map plays a crucial role in training our model. It allows the model to learn the correlation between the skeletal keypoints and the semantic information of the animal's body. By incorporating the segmentation map into the training process, the model can better understand the spatial relationships between different keypoints and improve the accuracy and robustness of pose estimation.

### Network training details

In our animal pose estimation tasks, we employed specific training configurations to optimize the performance of our models. The following training details were utilized. We trained the models for a total of 190 epochs. Additionally, we included 10 warm-up epochs at the beginning of the training process. The batch size used during training was set to 8. We utilized the *AdamW* optimizer, and the weight decay rate was set to 1e-4. We employed a warmup cosine decay schedule for the learning rate. Initially, the learning rate was warmed up from 1e-5 to 1e-3 over the warm-up epochs. Subsequently, the learning rate gradually decayed to 1e-5 using a cosine decay function. For optimizing the keypoint confidence heatmaps and location refinement maps, we utilized root square error (RMSE) as the loss function. RMSE measures the average squared difference between the predicted and ground truth key points, providing a measure of the accuracy of the model's predictions. Additionally, for training the low-resolution semantic segmentation map, we used sparse categorical cross-entropy loss, which is suitable for multi-class segmentation tasks. We early stop the training procedure when it reaches a plateau for 30 epochs according to validation loss. These training details were carefully chosen to ensure effective training and optimization of our models for single animal pose estimation. For data augmentation, we followed DeepLabCut augmentation strategy *Mathis et al., 2018* in training ADPT, and followed (*Pereira et al., 2022*) specifically for single fly dataset. The image inputs of ADPT were resized to a size that can be trained on the computer which was defined as 'global_scale' in configuration file. For mouse images, it was reduced to half of the original size. For monkey images, it was reduced to 0.8 of the original size. For macaque and fruit flies, there were no resizing, while for dogs, it was resized to 224×224 resolution. For homecage social mice images and marmoset images, there were no resizing.

The specific values and configurations may vary depending on the dataset, network architecture, and specific requirements of the task.

### Network implementation

We implemented ADPT in the Python programming language(python 3.9). We used tensorflow 2.9.1 for all deep learning models. We used imgaug for image and annotation augmentation. We

used OpenCV for video reading/writing and matplotlib for image reading. The hardware condition includes RTX4090 GPU, Intel 12,900 K CPU, Samsung 980 Pro hard disk, and 128 GB DDR5 memory. For comparison, we used DeepLabCut 2.2.1 with default configuration during training, in which 'global_scale' parameter was adjusted to match with ADPT resizing configuration. Similarly, SLEAP 1.2.9 was used with the baseline_medium_rf.single configuration, adjusting the 'input scaling' to align with ADPT's resizing configuration.

## Datasets

To comprehensively evaluate the robust performance of ADPT, we selected datasets consider factors such as skeletal complexity, body size, and background complexity. However, there exists no publicly available video dataset specifically designed for anti-drift evaluation. Therefore, we also collected behavioral video data involving mice and monkeys. We also provide code to transfer DeepLabCut format labeled dataset to our ADPT format dataset, which may allow users to make deeper study toward the past behavioral data. Code is available at https://github.com/tangguoling/ADPT/blob/main/data/dlc2adpt.py (*Tang, 2025*). Source data files have also been provided for *Figures 1–4 and 6–8*, details for accessing which are available at https://github.com/tangguoling/ADPT/blob/main/data/link.md.

## Mouse dataset

The mouse dataset is a customized single animal dataset collected by ourselves. We recorded a C57BL/6 mouse freely behaving in an open field from four different views. The dataset contained 440 labeled image in 1288×964 resolution across four different backgrounds and 11 individuals, 16 single mouse videos with the same resolution across 4 different individuals and four backgrounds. Each video spans 15 min.

## Monkey dataset

The monkey dataset is a customized single animal dataset collected by ourselves. We recorded a cynomolgus monkey freely behaving in behavioral cage. The dataset contained 3488 labeled image in 640×360 resolution across 8 different backgrounds and multiple individuals, and one specific 30 minutes video in which a monkey and people appeared simultaneously.

## Single fly dataset

The single fly dataset is a benchmark dataset used in animal pose estimation (*Pereira et al., 2022*). It contained 1500 manual labeled frames which was split into 1200 training, 150 validation, and 150 test frames. The fly in the dataset was annotated with 32-node skeleton. Source data files are available at https://github.com/jgraving/DeepPoseKit-Data/tree/master/datasets/fly.

## OpenMonkeyStudio Dataset

The OpenMonkeyStudio dataset is a macaque pose estimation dataset, containing 195,228 labeled frames with 13-node skeletons (*Bala et al., 2020*). we randomly selected 5000 images and resized them to 368×368 resolution to evaluate the performance of our methods. We randomly divided this selected dataset into a 40–60% training and validation split. Source data files are available at https://z.umn.edu/OMS_data_link.

## StanfordExtradataset

StanfordExtradataset is a large-scale dog dataset with 2D keypoint and silhouette annotations, containing 12,000 images of dogs with 24-node skeletons (*Biggs et al., 2020*). We randomly split the dataset into 85% training and 15% validation. Source data files are available at https://paperswithcode.com/dataset/stanfordextra.

## Mouse videos of different individuals

Video 1288×964 resolution across four different backgrounds and 10 individuals. Each video spans 15 min during which the first 12 min was used for training identity lrss while the rest was used for validation.

### Free-social mice video

A 1 min video in 1288×964 resolution of free-social mice.

### Homecage social mice dataset

The homecage social mice dataset is a customized animal dataset collected by ourselves. We recorded two markerless C57BL/6 mice freely behaving in a homecage from three different view. The dataset contained 1200 labeled images in 960×540 resolution across 3 different backgrounds and two paired individuals. Each video spans 10 min during which the first 1 min video was use for anti-drift performance evaluation. We manually annotated the position location in the 1 min video every 30 frames.

### Marmoset

Marmoset is a dataset released by multi-animal DeepLabCut for marmosets pose estimation, containing 5316 images of marmoset with 15-node skeletons *Lauer et al., 2022*. We resized the images to 368×368 resolution to evaluate the performance of our methods.

### Mix-up social animal dataset generation

To address the challenge of acquiring labeled datasets for multi-animal pose estimation, we introduce a novel data augmentation strategy. This strategy involves mixing up a background picture and two labeled frames from single animal videos predicted by single animal model, generating synthetic data with multiple animals. The process is illustrated in *Figure 6*, and the algorithm is detailed in Algorithm 1. Initially, we employ the ADPT model to predict keypoint position for two images originating from different videos, resulting in two frame annotation sets of keypoints. Using these frames and the corresponding background image (*Figure 6A*), we create a mix-up image, as shown in *Figure 6B*. We utilize two frame annotations to generate Mix-up annotation heatmaps. These heatmaps associate each keypoint with its corresponding location on the mix-up image, as shown in *Figure 6C*. For the augmented image as shown in *Figure 6D*, we generated augmented annotations as shown in *Figure 6E, F* represents augmented keypoints. Importantly, we leverage LRSS to distinguish between animals' identities, as indicated in the *Figure 6G*. Finally, we leverage body affinity fields (BAF) to match the body parts and identity, as indicated in the *Figure 6H* in which we set back as the center point.

### Body affinity fields

Inspired by PAF, we create Body Affinity Fields for associating body part to instance identity. Considering all individuals in frame, the pixel p-value at BAF map is defined as,

$$
\begin{cases}
(p_x - center_x, p_y - center_y), & \text{if } p \text{ on } body_i, \text{for } i \text{ in } instances \\
(0, 0), & \text{otherwise}
\end{cases}
\tag{2}
$$

where $p_x$ and $p_y$ represents pixel $p$'s location (x and y coordination), and $center_{x,y}$ represents the center location. Combining BAF and LRSS, we can infer pixels identities. We only used this map in social animal tracking.

---

**Algorithm 1. Generation of Mix-up Social Animal Data**

---

**Data**: video1, video2, backgrounds

**Result**: Mix-up frame, Mix-up annotation
1 **Tool**: ADPT; .
2 Select randomly;
3 $frame1 \in video1$;
4 $frame2 \in video2$;
5 $background \in backgrounds$;
6 Label frame;
7 $frame1\_annotation = ADPT(frame1)$;
8 $frame2\_annotation = ADPT(frame2)$;
9
$Mix\text{-}up\ annotation = \{frame1\_annotation, frame2\_annotation\}$
;
10 Mix up image;
11
$mouse1 = frame1[where((frame1 - background) >= delta)]$
;
12
$mouse2 = frame2[where((frame2 - background) >= delta\&mouse1 == 0)]$
;
13
$Mix\text{-}up\ frame = mouse1 + mouse2 + background[where((mouse1 + mouse2) == 0)]$
;

---

## Experiments for 10 mice identity tracking

In this experiment, we used videos featuring different identified mice, allocating 80% of the data for model training and the remaining 20% for accuracy validation. We configured the output channels of the model's LRSS to be 11 (1 background channel +10 identity channels). Finally, we determined the identity of mice in the image by analyzing the proportion of each category within the LRSS image. For data augmentation, random rotation (±30°), random pixel translation (x:[–100,100], y:[–30,15]) and random scale (0.9,1.1) were used in training ADPT.

Following metrics was used for identity determination:

$$identity = argmax(\sum p_{identity}) \tag{3}$$

where $p_{identity}$ represents pixel $p$ value at LRSS.

## Experiments for social mice tracking

In this experiment, we randomly selected two mice. We created a Mix-up Social Keypoint Dataset using individual videos of these mice and randomly captured background. We computed the BAF centered on the back of the mice. For the social interaction task, the LRSS channels of the model were set to 3 (one background channel and two identity channels), while two channels were introduced for the newly incorporated BAF (representing a two-dimensional vector). Random pixel translation (x:[–100,100], y:[–30,15]) was the only augmentation method used in training ADPT.

We trained the model on this mix-up dataset and used it to predict real social interaction videos of mice spanning 1 min. In practical application, we employed a bidirectional approach both bottom-up and top-down to ascertain mouse identities. Specifically, we utilized the BAF image to confirm the center position pointed by each pixel. Then, based on the identity information from LRSS corresponding to the center positions, we determined the identity information of each pixel (body pixels) to generate an identity map. Finally, by matching the location heatmap with the identity map, we calculated the posture information of the interacting animals.

Both manual verification and following metrics was used for evaluating identity exchange rate:

$$changerate@\alpha = \frac{1}{F}\sum_{i=2}^{F} \delta(\sqrt{(y_i - y_{i-1})^2} \leq \alpha) \tag{4}$$

where $y_i$ represents center location of each individual, and $\alpha$ represent drift distance threshold which was set as 75 pixels.

In our ten mice identity tracking and social mice tracking task, we trained the model for a total of 300 epochs with 10 warm-up epochs. We early stop the training procedure when it reaches a plateau for 30 epochs according to training loss. The batch size used during training was set to 8. Each epoch has 250 iterations for the first task and 50 iterations for the social task. To optimize the BAF maps, we utilized RMSE as the loss function.

### Evaluation metrics

To evaluate keypoint tracking drift, we use following metrics: for each keypoint,

$$drift@\alpha = \frac{1}{F}\sum_{i=2}^{F}\delta(\sqrt{(y_i - y_{i-1})^2} \leq \alpha) \tag{5}$$

where $F$ represents the total number of frames, $y_i$ represent a predicted keypoint position, $\alpha$ represent the drift distance threshold which was set as 50 pixels on mice, and 30 pixels on monkey, and $\delta$ is an indicator function that equals 1 when $\sqrt{(y_i - y_{i-1})^2} \leq \alpha$, and 0 otherwise.

$$failtodetect = \frac{1}{F}\sum_{i=1}^{F}\delta(y_{i,confidencescore} \leq 0.2) \tag{6}$$

where $y_{i,confidencescore}$ represents the confidence score of the predicted heatmap of i-th frames.

We used the following metrics for single animal pose estimation: PCK@0.15, RMSE, mAP

$$PCK@0.15 = \frac{1}{N}\sum_{i=1}^{N}\delta(d_i \leq 0.15 \cdot L_i) \tag{7}$$

where $N$ represents the total number of keypoints, $d_i$ is the Euclidean distance for the i-th keypoint, $L_i$ is the normalized limb scale associated with the i-th keypoint.

$$OKS = \frac{\sum_{i=1}^{N}\exp\left(-\frac{d_i^2}{2\alpha(2s)^2}\right)\delta_{v_i>0}}{\sum_{i=1}^{N}\delta_{v_i>0}} \tag{8}$$

where $\alpha$ is the bounding box area occupied by the GT instance, $v_i$ is a visibility flag for the i-th keypoint, and $s$ is the uncertainty factor(set to 0.025 for all measurements, the same as SLEAP)

$$AP@\alpha = \frac{1}{N}\sum_{i=1}^{N}\delta(OKS_i > \alpha) \tag{9}$$

where $\alpha$ represent the accuracy threshold.

$$mAP = \frac{1}{10}(AP@0.5 + AP@0.55 + AP@0.6 + ... + AP@0.95) \tag{10}$$

$$RMSE = \sqrt{\frac{1}{n}\sum_{i=1}^{n}(y_i - y_{\text{true},i})^2} \tag{11}$$

where $y_i$ represent a predicted keypoint position and $y_{\text{true},i}$ is its' ground truth.

## Acknowledgements

We acknowledge the effort from Wenhao Liu who recorded the mouse behavioral data and Professor Sen Yan's laboratory who recorded the monkey behavioral data. This work was supported in part by the National Natural Science Foundation of China (32222036 to PF W), Research Fund for International Senior Scientists (T2250710685 to PF W), STI2030-Major Projects (2021ZD0203900 to PFW), and Shenzhen Science and Technology Innovation Committee (2022410129 to QL). We thank ChatGPT for the English language editing of this paper.

## Additional information

### Funding

| Funder | Grant reference number | Author |
| --- | --- | --- |
| National Natural Science Foundation of China | 32222036 | Pengfei Wei |
| Research Fund for International Senior Scientists | T2250710685 | Pengfei Wei |
| STI2030-Major Projects | 2021ZD0203900 | Pengfei Wei |
| Shenzhen Science and Technology Innovation Program | 2022410129 | Quanying Liu |

The funders had no role in study design, data collection and interpretation, or the decision to submit the work for publication.

### Author contributions

Guoling Tang, Conceptualization, Resources, Data curation, Software, Formal analysis, Validation, Investigation, Visualization, Methodology, Writing – original draft, Writing – review and editing; Yaning Han, Conceptualization, Resources, Data curation, Validation, Investigation, Visualization, Methodology, Writing – original draft, Writing – review and editing; Xing Sun, Resources, Data curation, Software, Validation, Visualization, Methodology, Writing – review and editing; Ruonan Zhang, Software, Methodology; Ming-Hu Han, Resources, Supervision; Quanying Liu, Resources, Supervision, Funding acquisition, Validation, Visualization, Methodology, Writing – original draft, Writing – review and editing; Pengfei Wei, Conceptualization, Resources, Data curation, Supervision, Funding acquisition, Validation, Investigation, Visualization, Methodology, Writing – original draft, Project administration, Writing – review and editing

### Author ORCIDs

Guoling Tang ⓘ https://orcid.org/0009-0008-2318-2624
Yaning Han ⓘ https://orcid.org/0000-0002-1650-2262
Pengfei Wei ⓘ https://orcid.org/0000-0003-1845-8856

### Ethics

All experimental procedures of mice in this study were approved by Animal Care and Use Committees at the Shenzhen Institute of Advanced Technology, Chinese Academy of Sciences. And all experimental procedures of monkey adhered to the Guidelines for the Care and Use of Laboratory Animals established by Jinan University.

Reviewer #2 (Public review): https://doi.org/10.7554/eLife.95709.3.sa1
Author response https://doi.org/10.7554/eLife.95709.3.sa2

## Additional files

### Supplementary files

Supplementary file 1. Comparison among three methods on single fly dataset and OMS_Dataset.

MDAR checklist

### Data availability

All data generated or analysed during this study are included in the manuscript and supporting files; source data files have been provided for Figures 1, 2, 3, 4, 6, 7 and 8.

The following datasets were generated:

| Author(s) | Year | Dataset title | Dataset URL | Database and Identifier |
|---|---|---|---|---|
| Tang G | 2024 | Anti-drift pose tracker (ADPT): a transformer-based network for robust animal pose estimation cross-species (Part 1) | https://doi.org/10.5281/zenodo.10473121 | Zenodo, 10.5281/zenodo.10473121 |
| Tang G | 2024 | Anti-drift pose tracker (ADPT): A transformer-based network for robust animal pose estimation cross-species (Part 2) | https://doi.org/10.5281/zenodo.10473280 | Zenodo, 10.5281/zenodo.10473280 |
| Tang G | 2024 | Anti-drift pose tracker (ADPT): A transformer-based network for robust animal pose estimation cross-species (Part 3) | Https://doi.org/10.5281/zenodo.14218678 | Zenodo, 10.5281/zenodo.14218678 |
| Tang G | 2024 | ADPT-TOOLBOX Demonstration Mouse Video | https://doi.org/10.5281/zenodo.14566416 | Zenodo, 10.5281/zenodo.14566416 |

The following previously published datasets were used:

| Author(s) | Year | Dataset title | Dataset URL | Database and Identifier |
|---|---|---|---|---|
| Biggs B, Boyne O, Charles J, Fitzgibbon A, Cipolla R | 2020 | StanfordExtra | https://github.com/benjiebob/StanfordExtra | GitHub, StanfordExtra |
| Graving J, Chae D | 2019 | DeepPoseKit Data: example datasets for DeepPoseKit - Single fly dataset | https://github.com/jgraving/DeepPoseKit-Data/tree/master/datasets/fly | GitHub, DeepPoseKit-Data/tree/master/datasets/fly |
| Bala PC, Eisenreich BR, SBM Yoo, Hayden BY, Park HS, Zimmermann J | 2020 | OMS_Dataset | https://github.com/OpenMonkeyStudio/OMS_Data | GitHub, OpenMonkeyStudio/OMS_Data |

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

## Appendix 1

### Deep learning pose estimation

Pose estimation is a well-established computer vision task that has achieved significant advancements in human pose estimation. Traditional CNN-based algorithms for human pose estimation *Newell et al., 2016*; *Cao et al., 2021*; *Toshev and Szegedy, 2014*; *Chen et al., 2018*; *Wei et al., 2016*; *Insafutdinov et al., 2016*; *Sun et al., 2019* have been widely applied and have shown promising results. With the recent rise of transformer-based models, researchers have explored the use of transformers for human pose estimation (*Yang et al., 2021*; *Li et al., 2021*; *Xu and Zhang, 2022*; *Mao et al., 2021*), leading to improved accuracy and performance. At the same time, some of these works (*Newell et al., 2016*; *Wei et al., 2016*; *Insafutdinov et al., 2016*; *Xu and Zhang, 2022*) has also been extended to the field of animal pose estimation. Notably, keypoint detection methods typically employ two main approaches: heatmap-based and regression-based methods. Heatmap-based methods generate keypoint heatmaps, calculate the index of the maximum confidence score within these heatmaps, and obtain keypoint coordinates. Heatmap-based methods have the advantage of providing confidence scores, allowing researchers to gauge the reliability of each keypoint's estimate. However, they can be computationally intensive due to the generation of multiple heatmaps. Conversely, regression-based methods directly output keypoint coordinates from the model. Regression-based methods are often computationally efficient and can provide accurate results. However, they may lack the ability to express the confidence or uncertainty associated with each keypoint prediction, which heatmap-based methods can provide. The choice between these methods depends on the specific requirements of the pose estimation task.

In the domain of behavioral studies, specific estimation methods have been developed and widely used. Notable examples include DeepLabCut (*Mathis et al., 2018*), SLEAP *Pereira et al., 2022*, and DeepPoseKit (*Graving et al., 2019*). These methods have found extensive application in experimental animal pose estimation, where the estimated poses are used for quantifying and analyzing animal behavior. They are heatmap-based methods. DeepLabCut is a popular toolbox utilized for animal pose estimation, employing CNNs such as ResNets (*He et al., 2016*) or MobileNets (*Sandler et al., 2018*) that initial pretrained on ImageNet (*Russakovsky et al., 2015*) to accurately estimate animal poses. It has been widely adopted in various experimental settings, enabling researchers to track and analyze animal behavior with high precision. Similarly, SLEAP is another widely used tool for multi-animal pose estimation, leveraging U-NET (*Ronneberger et al., 2015*) liked CNN architectures to estimate poses and facilitate behavior analysis in animals. Additionally, DeepPoseKit is another notable software toolkit using Stacked DenseNet for behavioral animal pose estimation. The results of pose estimation serve as a critical component in quantifying and analyzing animal behavior. By accurately estimating animal poses, researchers can extract valuable insights into the kinematics (*Monsees et al., 2022*), dynamics (*Luxem et al., 2022*), and patterns of animal movements (*Huang et al., 2021*). This information further contributes to a better understanding of animal behavior, cognition, and underlying neural mechanisms.

According to literature report (*Pereira et al., 2022*), SLEAP and DeepLabCut have similar accuracy on a benchmark single-fly datasets (*Pereira et al., 2019*), with mean average precision scores(mAP) of 92.7% and 92.8%, respectively. Their accuracies are significantly higher than that of DeepPoseKit(86.4%). Additionally, SLEAP demonstrates the highest inference speed among the three tools. Therefore, currently, SLEAP and DeepLabCut are considered to have the best performance in freely behaving animal pose estimation. However, these methods are still limited by their robustness, which refers to the presence of uncertainty or noise interference in the estimated positions of keypoints due to the inherent limitations of the algorithms or noise in the image. For instance, the limited receptive fields of convolutional kernels may hinder their ability to capture the global dependencies within an image. This constraint can be particularly relevant in tasks that require modeling complex spatial relationships or long-range interactions. ADPT primarily aims to compare and improve upon these two methods.

In summary, various pose estimation methods, including DeepLabCut, SLEAP, and DeepPoseKit, have been developed and extensively employed in the field of experimental animal pose estimation. These methods leverage CNN-based models to estimate animal poses, enabling researchers to conduct detailed behavior quantification and analysis.

