## [Editor Report · eLife Assessment]

This **useful** study introduces a deep learning-based algorithm that tracks animal postures with reduced drift by incorporating transformers for more robust keypoint detection. The efficacy of this new algorithm for single-animal pose estimation was demonstrated through comparisons with two popular algorithms. The strength of evidence is **solid** but would benefit from consideration of issues in multi-animal tracking. This work will be of interest to those interested in animal behavior tracking.

---

## [Referee Report · Reviewer #2 (Public review)]

Summary:

The authors present a new model for animal pose estimation. The core feature they highlight is the model's stability compared to existing models in terms of keypoint drift. The authors test this model across a range of new and existing datasets. The authors also test the model with two mice in the same arena. For the single animal datasets the authors show a decrease in sudden jumps in keypoint detection and the number of undetected keypoints compared with DeepLabCut and SLEAP. Overall average accuracy, as measured by root mean squared error, generally shows generally similar but sometimes superior performance to DeepLabCut and better performance compared to SLEAP. The authors confusingly don't quantify the performance of pose estimation in the multi (two) animal case instead focusing on detecting individual identity. This multi-animal model is not compared with the model performance of the multi-animal mode of DeepLabCut or SLEAP.

Strengths:

The major strength of the paper is successfully demonstrating a model that is less likely to have incorrect large keypoint jumps compared to existing methods. As noted in the paper, this should lead to easier-to-interpret descriptions of pose and behavior to use in the context of a range of biological experimental workflows.

Weaknesses:

There are two main types of weaknesses in this paper. The first is a tendency to make unsubstantiated claims that suggest either model performance that is untested or misrepresents the presented data, or suggest excessively large gaps in current SOTA capabilities. One obvious example is in the abstract when the authors state ADPT "significantly outperforms the existing deep-learning methods, such as DeepLabCut, SLEAP, and DeepPoseKit." All tests in the rest of the paper, however, only discuss performance with DeepLabCut and SLEAP, not DeepPoseKit. At this point, there are many animal pose estimation models so it's fine they didn't compare against DeepPoseKit, but they shouldn't act like they did. Similar odd presentation of results are statements like "Our method exhibited an impressive prediction speed of 90{plus minus}4 frames per second (fps), faster than DeepLabCut (44{plus minus}2 fps) and equivalent to SLEAP (106{plus minus}4 fps)." Why is 90{plus minus}4 fps considered "equivalent to SLEAP (106{plus minus}4 fps)" and not slower? I agree they are similar but they are not the same. The paper's point of view of what is "equivalent" changes when describing how "On the single-fly dataset, ADPT excelled with an average mAP of 92.83%, surpassing both DeepLabCut and SLEAP (Figure 5B)" When one looks at Figure 5B, however, ADPT and DeepLabCut look identical. Beyond this, oddly only ADPT has uncertainty bars (no mention of what uncertainty is being quantified) and in fact, the bars overlap with the values corresponding to SLEAP and DeepPoseKit. In terms of making claims that seem to stretch the gaps in the current state of the field, the paper makes some seemingly odd and uncited statements like "Concerns about the safety of deep learning have largely limited the application of deep learning-based tools in behavioral analysis and slowed down the development of ethology" and "So far, deep learning pose estimation has not achieved the reliability of classical kinematic gait analysis" without specifying which classical gait analysis is being referred to. Certainly, existing tools like DeepLabCut and SLEAP are already widely cited and used for research.

The other main weakness in the paper is the validation of the multi-animal pose estimation. The core point of the paper is pose estimation and anti-drift performance and yet there is no validation of either of these things relating to multi-animal video. All that is quantified is the ability to track individual identity with a relatively limited dataset of 10 mice IDs with only two in the same arena (and see note about train and validation splits below). While individual tracking is an important task, that literature is not engaged with (i.e. papers like Walter and Couzin, eLife, 2021: https://doi.org/10.7554/eLife.64000) and the results in this paper aren't novel compared to that field's state of the art. On the other hand, while multi-animal pose estimation is also an important problem the paper doesn't engage with those results either. The two methods already used for comparison in the paper, SLEAP and DeepPoseKit, already have multi-animal modes and multi-animal annotated datasets but none of that is tested or engaged with in the paper. The paper notes many existing approaches are two-step methods, but, for practitioners, the difference is not enough to warrant a lack of comparison. The authors state that "The evaluation of our social tracking capability was performed by visualizing the predicted video data (see supplement Videos 3 and 4)." While the authors report success maintaining mouse ID, when one actually watches the key points in the video of the two mice (only a single minute was used for validation) the pose estimation is relatively poor with tails rarely being detected and many pose issues when the mice get close to each other.

Finally, particularly in the methods section, there were a number of places where what was actually done wasn't clear. For example in describing the network architecture, the authors say "Subsequently, network separately process these features in three branches, compute features at scale of one-fourth, one-eight and one-sixteenth, and generate one-eight scale features using convolution layer or deconvolution layer." Does only the one-eight branch have deconvolution or do the other branches also? Similarly, for the speed test, the authors say "Here we evaluate the inference speed of ADPT. We compared it with DeepLabCut and SLEAP on mouse videos at 1288 x 964 resolution", but in the methods section they say "The image inputs of ADPT were resized to a size that can be trained on the computer. For mouse images, it was reduced to half of the original size." Were different image sizes used for training and validation? Or Did ADPT not use 1288 x 964 resolution images as input which would obviously have major implications for the speed comparison? Similarly, for the individual ID experiments, the authors say "In this experiment, we used videos featuring different identified mice, allocating 80% of the data for model training and the remaining 20% for accuracy validation." Were frames from each video randomly assigned to the training or validation sets? Frames from the same video are very correlated (two frames could be just 1/30th of a second different from each other), and so if training and validation frames are interspersed with each other validation performance doesn't indicate much about performance on more realistic use cases (i.e. using models trained during the first part of an experiment to maintain ids throughout the rest of it.)

Editors' note: None of the original reviewers responded to our request to re-review the manuscript. The attached assessment statement is the editor's best attempt at assessing the extent to which the authors addressed the outstanding concerns from the previous round of revisions.

---

## [Author Response]

The following is the authors’ response to the original reviews.

**eLife Assessment**
This study introduces a useful deep learning-based algorithm that tracks animal postures with reduced drift by incorporating transformers for more robust keypoint detection. The efficacy of this new algorithm for single-animal pose estimation was demonstrated through comparisons with two popular algorithms. However, the analysis is incomplete and would benefit from comparisons with other state-of-the-art methods and consideration of multi-animal tracking.

First, we would like to express our gratitude to the eLife editors and reviewers for their thorough evaluation of our manuscript. ADPT aims to improve the accuracy of body point detection and tracking in animal behavior, facilitating more refined behavioral analyses. The insights provided by the reviewers have greatly enhanced the quality of our work, and we have addressed their comments point-by-point.

In this revision, we have included additional quantitative comparisons of multi-animal tracking capabilities between ADPT and other state-of-the-art methods. Specifically, we have added evaluations involving homecage social mice and marmosets to comprehensively showcase ADPT’s advantages from various perspectives. This additional analysis will help readers better understand how ADPT effectively overcomes point drift and expands its applicability in the field.

**Reviewer #1:**
In this paper, the authors introduce a new deep learning-based algorithm for tracking animal poses, especially in minimizing drift effects. The algorithm's performance was validated by comparing it with two other popular algorithms, DeepLabCut and LEAP.The accessibility of this tool for biological research is not clearly addressed, despite its potential usefulness. Researchers in biology often have limited expertise in deep learning training, deployment, and prediction. A detailed, step-by-step user guide is crucial, especially for applications in biological studies.

We appreciate the reviewers' acknowledgment of our work. While ADPT demonstrates superior performance compared to DeepLabCut and SLEAP, we recognize that the absence of a user-friendly interface may hinder its broader application, particularly for users with a background solely in biology. In this revision, we have enhanced the command-line version of the user tutorial to provide a clear, step-by-step guide. Additionally, we have developed a simple graphical user interface (GUI) to further support users who may not have expertise in deep learning, thereby making ADPT more accessible for biological research.

The proposed algorithm focuses on tracking and is compared with DLC and LEAP, which are more adept at detection rather than tracking.

In the field of animal pose estimation, the distinction between detection and tracking is often blurred. For instance, the title of the paper "SLEAP: A deep learning system for multi-animal pose tracking" refers to "tracking," while "detection" is characterized as "pose estimation" in the body text. Similarly, "Multi-animal pose estimation, identification, and tracking with DeepLabCut" uses "tracking" in the title, yet "detection" is also mentioned in the pose estimation section. We acknowledge that referencing these articles may have contributed to potential confusion.

To address this, we have clarified the distinction between "tracking" and "detection" Results section under " Anti-drift pose tracker." (see lines 118-119). In this paper, we now explicitly use “track” to refer to the tracking of all body points or poses of an individual, and “detect” for specific keypoints.

**Reviewer #1 recommendations:**
(1) DLC and LEAP are mainly good in detection, not tracking. The authors should compare their ADPT algorithm with idtracker.ai, ByteTrack, and other advanced tracking algorithms, including recent track-anything algorithms.(2) DeepPoseKit is outdated and no longer maintained; a comparison with the T-REX algorithm would be more appropriate.

We appreciate the reviewer's suggestion for a more comprehensive comparison and acknowledge the importance of including these advanced tracking algorithms. However, we have not yet found suitable publicly available datasets for such comparative testing. We appreciate this insight and will consider incorporating T-REX into future comparisons.

(3) The authors primarily compared their performance using custom data. A systematic comparison with published data, such as the dataset reported in the paper "Multi-animal pose estimation, identification, and tracking with DeepLabCut," is necessary. A detailed comparison of the performances between ADPT and DLC is required.

In the previous version of our manuscript, we included the SLEAP single-fly public dataset and the OMS_dataset from OpenMonkeyStudio for performance comparisons. We recognize that these datasets were not comprehensive. In this revision, we have added the marmoset dataset from "Multi-animal pose estimation, identification, and tracking with DeepLabCut" and a customized homecage social mice dataset to enhance our comparative analysis of multi-animal pose estimation performance. Our comprehensive comparison reveals that ADPT outperforms both DLC and SLEAP, as discussed in the Results section under "ADPT can be adapted for end-to-end pose estimation and identification of freely social animals.". (Figure 1, see lines 303-332)

(4) Given the focus on biological studies, an easy-to-use interface and introduction are essential.

In this revision, we have not only developed a GUI for ADPT but also included a more detailed tutorial. This can be accessed at https://github.com/tangguoling/ADPT-TOOLBOX

**Reviewer #2:**
The authors present a new model for animal pose estimation. The core feature they highlight is the model's stability compared to existing models in terms of keypoint drift. The authors test this model across a range of new and existing datasets. The authors also test the model with two mice in the same arena. For the single animal datasets the authors show a decrease in sudden jumps in keypoint detection and the number of undetected keypoints compared with DeepLabCut and SLEAP. Overall average accuracy, as measured by root mean squared error, generally shows similar but sometimes superior performance to DeepLabCut and better performance compared to SLEAP. The authors confusingly don't quantify the performance of pose estimation in the multi (two) animal case instead focusing on detecting individual identity. This multi-animal model is not compared with the model performance of the multi-animal mode of DeepLabCut or SLEAP.

We appreciate the reviewer's thoughtful assessment of our manuscript. Our study focuses on addressing the issue of keypoint drift prevalent in animal pose estimation methods like DeepLabCut and SLEAP. During the model design process, we discovered that the structure of our model also enhances performance in identifying multiple animals. Consequently, we included some results related to multi-animal identity recognition in our manuscript.

In recent developments, we are working to broaden the applicability of ADPT for multi-animal pose estimation and identity recognition. Given that our manuscript emphasizes pose estimation, we have added a comparison of anti-drift performance in multi-animal scenarios in this revision. This quantifies ADPT's capability to mitigate drift in multi-animal pose estimation.

Using our custom Homecage social mice dataset, we compared ADPT with DeepLabCut and SLEAP. The results indicate that ADPT achieves more accurate anti-drift pose estimation for two mice, with superior keypoint detection accuracy. Furthermore, we also evaluated pose estimation accuracy on the publicly available marmoset dataset, where ADPT outperformed both DeepLabCut and SLEAP. These findings are discussed in the Results section under "ADPT can be adapted for end-to-end pose estimation and identification of freely social animals."

The first is a tendency to make unsubstantiated claims that suggest either model performance that is untested or misrepresents the presented data, or suggest excessively large gaps in current SOTA capabilities. One obvious example is in the abstract when the authors state ADPT "significantly outperforms the existing deep-learning methods, such as DeepLabCut, SLEAP, and DeepPoseKit." All tests in the rest of the paper, however, only discuss performance with DeepLabCut and SLEAP, not DeepPoseKit. At this point, there are many animal pose estimation models so it's fine they didn't compare against DeepPoseKit, but they shouldn't act like they did.

We appreciate the reviewer's feedback regarding unsubstantiated claims in our manuscript. Upon careful review, we acknowledge that our previous revisions inadvertently included statements that may misrepresent our model's performance. In particular, we have revised the abstract to eliminate the mention of DeepPoseKit, as our comparisons focused exclusively on DeepLabCut and SLEAP.

In addition to this correction, we have thoroughly reviewed the entire manuscript to address other instances of ambiguity and ensure that our claims are well-supported by the data presented. Thank you for bringing this to our attention; we are committed to maintaining the integrity of our claims throughout the paper.

In terms of making claims that seem to stretch the gaps in the current state of the field, the paper makes some seemingly odd and uncited statements like "Concerns about the safety of deep learning have largely limited the application of deep learning-based tools in behavioral analysis and slowed down the development of ethology" and "So far, deep learning pose estimation has not achieved the reliability of classical kinematic gait analysis" without specifying which classical gait analysis is being referred to. Certainly, existing tools like DeepLabCut and SLEAP are already widely cited and used for research.

In this revision, we have carefully reviewed the entire manuscript and addressed the instances of seemingly odd and unsubstantiated claims. Specifically, we have revised the statements "largely limited" to "limited" to ensure accuracy and clarity. Additionally, we thoroughly reviewed the citation list to ensure proper attribution, incorporating references such as "A deep learning-based toolbox for Automated Limb Motion Analysis (ALMA) in murine models of neurological disorders" to better substantiate our claims and provide a clearer context.

We have also added an additional section to comprehensively discuss the applications of widely-used tools like DeepLabCut and SLEAP in behavioral research. This new section elaborates on the challenges and limitations researchers encounter when applying these methods, highlighting both their significant contributions and the areas where improvements are still needed.

The other main weakness in the paper is the validation of the multi-animal pose estimation. The core point of the paper is pose estimation and anti-drift performance and yet there is no validation of either of these things relating to multi-animal video. All that is quantified is the ability to track individual identity with a relatively limited dataset of 10 mice IDs with only two in the same arena (and see note about train and validation splits below). While individual tracking is an important task, that literature is not engaged with (i.e. papers like Walter and Couzin, eLife, 2021: https://doi.org/10.7554/eLife.64000) and the results in this paper aren't novel compared to that field's state of the art. On the other hand, while multi-animal pose estimation is also an important problem the paper doesn't engage with those results either. The two methods already used for comparison in the paper, SLEAP and DeepPoseKit, already have multi-animal models and multi-animal annotated datasets but none of that is tested or engaged with in the paper. The paper notes many existing approaches are two-step methods, but, for practitioners, the difference is not enough to warrant a lack of comparison.

We appreciate the reviewer's insights regarding the validation of multi-animal pose estimation in our paper. While our primary focus has been on pose estimation and anti-drift performance, we recognize the importance of validating these aspects within the context of multi-animal videos.

In this revision, we have included a comparison of ADPT's anti-drift performance in multi-animal pose estimation, utilizing our custom Homecage social mouse dataset (Figure 1A). Our findings indicate that ADPT achieves more accurate pose estimation for two mice while significantly reducing keypoint drift, outperforming both DeepLabCut and SLEAP. (see lines 311-322). We trained each model three times, and this figure presents the results from one of those training sessions. We calculated the average RMSE between predictions and manual labels, demonstrating that ADPT achieved an average RMSE of 15.8 ± 0.59 pixels, while DeepLabCut (DLC) and SLEAP recorded RMSEs of 113.19 ± 42.75 pixels and 94.76 ± 1.95 pixels, respectively (Figure 1C). ADPT achieved an accuracy of 6.35 ± 0.14 pixels based on the DLC evaluation metric across all body parts of the mice, while DLC reached 7.49 ± 0.2 pixels (Figure 1D). ADPT achieved 8.33 ± 0.19 pixels using the SLEAP evaluation Metric across all body parts of the mice, compared to SLEAP’s 9.82 ± 0.57 pixels (Figure 1E).

Furthermore, we have conducted pose estimation accuracy evaluations on the publicly available marmoset dataset from DeepLabCut, where ADPT also demonstrated superior performance compared to DeepLabCut and SLEAP. These results can be found in the "ADPT can be adapted for end-to-end pose estimation and identification of freely social animals" section of the Results. (see lines 323-329)

We acknowledge the existing literature on multi-animal tracking, such as the work by Walter and Couzin (2021). While individual tracking is crucial, our primary focus lies in the effective tracking of animal poses and minimizing drift during this process. This dual emphasis on pose tracking and anti-drift performance distinguishes our work and aligns with ongoing advancements in the field. Engaging with relevant literature, highlights the importance of contextualizing our results within the broader tracking literature, demonstrating that while our findings may overlap with existing methods, the unique focus on improving tracking stability and reducing drift presents valuable contributions to the field. Thank you for your valuable feedback, which has helped us improve the robustness of our manuscript.

The authors state that "The evaluation of our social tracking capability was performed by visualizing the predicted video data (see supplement Videos 3 and 4)." While the authors report success maintaining mouse ID, when one actually watches the key points in the video of the two mice (only a single minute was used for validation) the pose estimation is relatively poor with tails rarely being detected and many pose issues when the mice get close to each other.

We acknowledge that there are indeed challenges in pose estimation, particularly when the two mice get close to each other, leading to tracking failures and infrequent detection of tails in the predicted videos. The reasons for these issues can be summarized as follows:

Lack of Training Data from Real Social Scenarios: The training data used for the social tracking assessment were primarily derived from the Mix-up Social Animal Dataset, which does not fully capture the complexities of real social interactions. In future work, we plan to incorporate a blend of real social data and the Mix-up data for model training. Specifically, we aim to annotate images where two animals are in close proximity or interacting to enhance the model's understanding of genuine social behaviors.

Challenges in Tail Tracking in Social Contexts: Tracking the tails of mice in social situations remains a significant challenge. To validate this, we have added an assessment of tracking performance in real social settings using homecage data. Our findings indicate that using annotated data from real environments significantly improves tail tracking accuracy, as demonstrated in the supplementary video.

We appreciate your feedback, which highlights critical areas for improvement in our model.

Finally, particularly in the methods section, there were a number of places where what was actually done wasn't clear.

We have carefully reviewed and revised the corresponding parts to clarify the previously incomprehensible statements. Thank you for your valuable feedback, which has helped enhance the clarity of our methods.

For example in describing the network architecture, the authors say "Subsequently, network separately process these features in three branches, compute features at scale of one-fourth, one-eight and one-sixteenth, and generate one-eight scale features using convolution layer or deconvolution layer." Does only the one-eight branch have deconvolution or do the other branches also?

We apologize for the confusion this has caused. Upon reviewing our manuscript, we identified an error in the diagram. In the revised version, we have clarified that the model samples feature maps at multiple resolutions and ultimately integrates them at the 1/8 resolution for feature fusion. Specifically, the 1/4 feature map from ResNet50's stack 2 is processed through max-pooling and convolution to generate a 1/8 feature map. Additionally, the 1/4 feature map from ResNet50's stack 2 is also transformed into a 1/8 feature map using a convolution operation with a stride of 2. Finally, both the input and output of the transformer are at the 1/16 resolution, which can be trained on a 2080Ti GPU. The 1/16 feature map is then upsampled to produce the final 1/8 feature map. We have updated the manuscript to reflect these changes, and we also modified the model architecture diagram for better clarity.

Similarly, for the speed test, the authors say "Here we evaluate the inference speed of ADPT. We compared it with DeepLabCut and SLEAP on mouse videos at 1288 x 964 resolution", but in the methods section they say "The image inputs of ADPT were resized to a size that can be trained on the computer. For mouse images, it was reduced to half of the original size." Were different image sizes used for training and validation? Or Did ADPT not use 1288 x 964 resolution images as input which would obviously have major implications for the speed comparison?

For our inference speed evaluation, all models, including ADPT, used images with a resolution of 1288 x 964. In ADPT's processing pipeline, the first layer is a resizing layer designed to compress the images to a scale determined by the global scale parameter. For the mouse images, we set the global scale to 0.5, allowing our GPU to handle the data at that resolution during transformer training.

We recorded the time taken by ADPT to process the entire 15-minute mouse video, which included the time taken for the resizing operation, and subsequently calculated the frames per second (FPS). We have clarified this process in the manuscript, particularly in the "Network Architecture" section, where we specify: "Initially, ADPT will resize the images to a390 scale (a hyperparameter, consistent with the global scale in the DLC configuration)."

Similarly, for the individual ID experiments, the authors say "In this experiment, we used videos featuring different identified mice, allocating 80% of the data for model training and the remaining 20% for accuracy validation." Were frames from each video randomly assigned to the training or validation sets? Frames from the same video are very correlated (two frames could be just 1/30th of a second different from each other), and so if training and validation frames are interspersed with each other validation performance doesn't indicate much about performance on more realistic use cases (i.e. using models trained during the first part of an experiment to maintain ids throughout the rest of it.)

In our study, we actually utilized the first 80% of frames from each video for model training and the remaining 20% for testing the model's ID tracking accuracy. We have revised the relevant description in the manuscript to clarify this process. The updated description can be found in the "Datasets" section under "Mouse Videos of Different Individuals."